# Telomere Dynamics in the Diploid and Triploid Rainbow Trout (*Oncorhynchus mykiss*) Assessed by Q-FISH Analysis

**DOI:** 10.3390/genes11070786

**Published:** 2020-07-13

**Authors:** Ligia Panasiak, Stefan Dobosz, Konrad Ocalewicz

**Affiliations:** 1Department of Marine Biology and Ecology, Institute of Oceanography, Faculty of Oceanography and Geography, University of Gdansk, M. Piłsudskiego 46 Av, 81378 Gdynia, Poland; ligia.panasiak@phdstud.ug.edu.pl; 2Department of Salmonid Research, Inland Fisheries Institute in Olsztyn, Rutki, 10719 Olsztyn, Poland; s.dobosz@infish.com.pl

**Keywords:** aging, growth, telomere attrition

## Abstract

Changes of telomere length with age were assessed in diploid and triploid rainbow trout *(Oncorhynchus mykiss*) females in the cross-sectional study using Q-FISH technique. Triploid trout as sterile do not invest an energy in gametogenesis and continue to grow, whereas fertile diploid individuals suffer from declines in growth and survival during sexual maturation. However, triploid and diploid specimens exhibited similar patterns of telomere dynamics. Telomere length in the embryos, larvae and one-year-old juveniles did not change significantly. In the second year after hatching, subadults exhibited substantially shortened telomeres, while significant increase of the telomere length was reported in the three-year-old adults. On the other hand, correlation between telomere length and body size was observed in the triploid, but not in the diploid rainbow trout. Telomere shortening observed in two-year-old subadults may have been associated with the premature period of the fast growth in rainbow trout. Similar pattern of the telomere dynamics reported in the fertile diploids and sterile triploids indicated processes related to reproduction did not affect telomere dynamics in this species. Unexpected increase of the telomere length reported during the third year of life confirmed that in rainbow trout telomeric DNA shortens and lengthens, depending on the developmental stage.

## 1. Introduction

Telomeres are nucleoprotein complexes at the ends of eukaryotic chromosomes. In all vertebrates studied to date, the DNA component of the telomeres contains tandemly repeated G-rich hexanucleotide sequence (TTAGGG/CCCTAA)_n_ [1]. Telomeres protect chromosomes from end-to-end fusions and degradation, guarantee their complete replication and allow DNA repair machinery to distinguish natural chromosomal ends from the ends that appear in the course of breakage events [2,3]. As the DNA polymerase is not able to replicate ends of linear chromosomes (“end replication problem”) telomeric DNA shorten with every cell division [4]. Moreover, attrition of telomeres is accelerated by the oxidative stress associated with increased production of reactive oxygen species (ROS) [5]. ROS are generated during aerobic metabolism and ATP production in mitochondria. Moreover, ROS are formed in the course of exposure to the UV light, ionizing radiation and xenobiotics [6]. Rich in guanine telomeric DNA is particularly susceptible to damage from ROS and many experiments confirmed that oxidative stress increases incidences of the DNA strand breaks leading to loss of distal telomere fragments [7]. The process of telomere loss may be compensated mainly by the telomerase, an enzyme whose catalytic protein subunit (TERT, telomerase reverse transcriptase) adds telomeric DNA repeats to the end of telomeres using as a template an integral RNA component (TR, telomerase RNA). In contrast to humans, where telomerase expression in the adults is restricted to germ line cells, stem cells and tumors, expression of telomerase in various ectotherm species has been detected in many tissues irrespective of age [8,9,10].

Consistent decline of the telomere length with age has usually been observed in the endotherms [11]. In ectotherms results concerning relationship between age and telomere length are not that equivocal [12]. Age-related telomere shortening has been confirmed in alligators (*Alligator mississippiensis*) [13] and garter snakes (*Thamnophis elegans*) [14], but not in the leatherback turtle (*Dermochelys coriacea*) [15]. In fish, age-dependent decline of telomeres has been reported in some strains of Japanese medaka (*Oryzias latipes*) [16] and killifish (*Nothobranchius furzeri*) [17], but no gradual attrition of telomeric DNA with age was evidenced in the European sea bass (*Dicentrarchus labrax*) [18] or Atlantic silverside (*Menidia menidia*) [19]. In turn, in zebrafish (*Danio rerio*), the telomere length increases from larvae to the adult stage and shorten significantly in the aged individuals [20].

Somatic growth and reproduction are energetically costly and generate large amount of ROS and both processes have been found to affect dynamics of telomere attrition [21]. Taking into account that energetic investments in the growth and reproduction vary between males and females, intersexual differences in the telomere length should not be surprising. Indeed, in many species studied under this regard, females live longer than males and also have longer telomeres [22]. Recent experiments show that due to the estrogens, mitochondria from females produces less reactive oxygen species than males [23]. Moreover, in the egg producing animals, vitellogenin that is an egg yolk precursor has also been found to act as an antioxidant [24].

Although, processes related to reproduction are likely to matter in the context of telomere length, little attention has been devoted to analyzing telomere dynamics in the triploid sterile fish. The additional set of chromosomes causes cytogenetic incompatibility, which makes most of the triploids unable to reproduce. Triploid fish do not invest an energy in gametogenesis and continue to grow whereas normal diploid specimens suffer from declines in growth and survival during sexual maturation [25,26]. Spontaneous occurrence of triploids was confirmed in several fish species [27,28,29]. Moreover, methods of artificial triploidization has been elaborated for many commercially important aquaculture fishes [30]. Currently, triploids are used in the production of rainbow trout (*Oncorhynchus mykiss* Walbaum, 1792)—one of the most important salmonid fish species in world aquaculture [25]. Although, rainbow trout was the first fish with confirmed multitissue activity of telomerase irrespective of the individual age [8], to date, no information concerning alterations of the telomere length with age has been provided in this species. Therefore, the main goal of the present research was to examine changes of the telomere length in fertile diploid and sterile triploid rainbow trout sampled at different developmental stages from embryogenesis to adulthood. To assess the dynamics of telomeres, sensitive and specific quantitative fluorescence in situ hybridization (Q-FISH) method has been utilized [31,32]. This technique includes the application of fluorescein-conjugated peptide nucleic acid (PNA) telomere probe that stains telomeres proportionally to their length and the software that enables capturing and quantification of the fluorescent signals. Q-FISH has been successfully applied to interphase nuclei to quantify change in the telomere length with age in several species including fish [20].

## 2. Materials and Method

This study was carried out in strict accordance with the recommendations in the Polish ACT of 21 January 2005 of Animal Experiments (Dz. U. of. 2005 No 33, item 289). The protocol was approved by the Local Ethical Committee for the Experiments on Animals in Bydgoszcz.

### 2.1. Fish Origin

Embryos at 16 day post-fertilization (dpf), larvae (28 dpf), juvenile (one-year-old), subadult (two-year-old) and adult (three-year-old) females from the diploid and triploid stocks of rainbow trout from Rutki strain, reared in the Department of Salmonid Research (DSR), Inland Fisheries Institute in Olsztyn (IFI), Rutki, Poland were sampled for examination of the telomere length dynamics. No fish older than three years were available. Triploid stocks from the Department of Salmonid Research were produced using standard procedure including application of high hydrostatic pressure (HHP) shock (9000 psi/3 min.) to the fertilized eggs 35 min. after insemination (Dobosz, personal communication).

### 2.2. Egg Incubation and Fish Rearing

Diploid and triploid rainbow trout were reared separately, but under the same husbandry and environmental conditions. The hatchery and the outdoor grow-out facilities were supplied with water from Radunia River that average temperature was 11 °C (range: winter 0–4 °C, spring 4–16 °C, summer 15–22 °C, autumn 5–14 °C). Eggs were incubated in the conventional flow through incubators. After hatching, larvae at the swim-up stage were transferred to the plastic tanks (1 m^3^) where they were reared during the first year of life. Within their second and third year of life, diploid and triploid fish were kept outdoor in the rectangular (10 m^3^) and rotational concrete ponds (56 m^3^), respectively. Fish were fed daily and feeding rates were adjusted to the growth and daily temperatures. When water temperature was between 18 and 20 °C feeding was reduced to half. Feeding was fully stopped when water temperature was higher than 20 °C. During the entire experiment, fish were held under natural light conditions.

### 2.3. Preparation of Interphase Spreads

In the present research, eight embryos, eight larvae and five juveniles, six subadults and five adults randomly chosen from each, diploid and triploid stock were examined. The sample size was determined to meet requirements for the statistical analysis, based on the previous studies in fish [20]. All sampled trout individuals were sacrificed by an overdose of propiscin (*Etomidatum*, IFI (Inland Fisheries Institute, Olsztyn, Poland), weighed and their length measured. In the diploid and triploid juveniles, subadults and adults, development of reproductive organs and presence of oocytes were assessed macroscopically after fish dissection.

Suspensions of cells from embryos and larvae were prepared with the modified method described by Polonis et al. [33]. Briefly, full larvae and embryos that had been gently removed from the eggs were incubated in 0.075-M KCl solution for 30 min at room temperature and then transferred to the tubes with 50 µL of fixative (methanol: acetic acid, 3:1), which was changed twice. The tissue was left in the fixative for 20 min. Heads of the fixed embryos and larvae were cut off and macerated with scissors and dissociated by pipetting in 60 µL of methanol–acetic acid (2:1) to obtain homogeneous cell suspensions. Single drops of cell suspension from each individual were dropped on the microscope slides and left to dry at room temperature for 24 h.

The somatic cells of juveniles, subadults and adults were prepared from the head kidney tissue. Head kidney is a primary fish lymphoid and endocrine organ usually used for preparation of metaphase and interphase spreads for the cytogenetic analysis including fluorescence in situ hybridization (FISH) with PNA (peptide nucleic acid) telomere probe [34,35]. Portions of the head kidney were placed in tubes with 5 mL of KCl (0.075 M), homogenized and left for 40 min at room temperature. Next, 10 drops of freshly prepared ice cold fixative (methanol: acetic acid, 3:1) were added. After 2 min, the tubes were filled with the fixative up to 10 mL and centrifuged at 160× *g* for 10 min. Then, the supernatant was tossed out and fresh fixative added. Samples were kept at −20 °C for 30 min. The fixative was changed three times. After the final centrifugation, the supernatant was replaced by a freshly prepared fixative (1–2 mL) and such prepared cell suspensions were transferred to plastic tubes and stored at −20 °C for the further use. Microscope slides were prepared by placing one drop of the cell suspension from a height of 20–30 cm on a slide and left to dry.

### 2.4. Interphase Quantitative Fluorescence In Situ Hybridization (Q-FISH)

Microscope slides with rainbow trout cells were prepared the day before Q-FISH procedure and kept at the room temperature. Telomeric DNA repeats were detected by FISH, using a Telomere PNA (peptide nucleic acid) FISH Kit/FITC (DAKO, Glostrup, Denmark) according to the manufacturer’s protocol. Chromosomal DNA was denatured at 85 °C for 5 min under the cover slip in the presence of the PNA probe. During hybridization, microscope slides were left in the dark at room temperature for 1 h. Fifteen minutes before microscopic analysis cells were counterstained with DAPI in Vectashield antifade solution (Vector Laboratories, Burlingame, CA, USA). FISH was performed on two slides from each studied specimen.

Each slide was scanned with the ASI system, HiFISH-SpotScan module (Applied Spectral Imaging, Yokne’am Illit, Israel) with a dedicated 5 M CMOS camera connected to a fluorescent BX53 Olympus microscope. Ten (10) fields of view (frames) were captured under 100× magnification with DAPI and FITC filters in the multiple focal planes to ensure all signals will be in focus for the correct analysis. SpotScan automatically detected all cells in the scanned region and the telomere fluorescent signals within them. Quantitative analysis of the telomere fluorescence intensity was performed on the classified cells. Average telomere fluorescence intensities and standard deviations were automatically reported per 100–200 cells from the scanned regions. Detailed data concerning particular frames are available on request due to privacy/ethical restrictions. The optical and software setup was kept identical along the scan of the slide and between slides. Note that the intensity values were given in the arbitrary units, as acceptable in such cases. Any conclusion was derived by comparison between paralleled experiments.

### 2.5. Statistical Analysis

Data were analyzed using R software version 3.5.3 (11 March 2019). Normal distribution was tested by Shapiro–Wilk test. Correlations between telomere length-related fluorescence and fish body weight (1) and length (2) were assessed using Pearson or Spearman test, according to the data distribution.

Since age, ploidy, body size (length and weight) and interactions between these variables may affect telomere length, results were analyzed using analysis of covariance (ANCOVA) and general linear model (GLM) was provided. As body length and weight are correlated, two separate ANCOVA were performed. Afterward, ANOVA post hoc Tukey’s test was applied.

## 3. Results

### 3.1. Body Weight and Length and Gonadal Development

Body length and weight of the examined specimens increased with age, what confirmed that rainbow trout grow continuously throughout life. Diploid embryos, larvae, juveniles and subadults exhibited significantly lower body length than triploids while adults, irrespective of the ploidy, showed similar length (*p* < 0.05). In turn, triploid larvae, juveniles and two-year-old trout were substantially heavier than their diploid counterparts (Table 1). Weight of embryos and adults from diploid and triploid stocks did not differ substantially (*p* > 0.05).

Ovaries were macroscopically observed only in the diploid juveniles and subadults. In diploid adult females, oocytes were found in the body cavity. In the triploid individuals neither oocytes nor properly developed ovaries were observed what confirmed sterility of these fish.

### 3.2. Dynamics of Telomere Length in Rainbow Trout

Q-FISH on the rainbow trout interphase diploid and triploid cells was successfully applied to follow changes of the telomere length with age and body growth (Figure 1). Mean values of the telomere length-related intensity of fluorescence for individuals from each age category and ploidy are provided (Appendix A). In the diploid rainbow trout embryos, larvae and juveniles, telomere length-related fluorescence presented in the fluorescent arbitrary units ×10 equaled 24.94 ± 6.07 (mean ± SD), 26.61 ± 4.91 and 21.73 ± 5.06, respectively. Substantially shortened telomere length was observed only in the subadults (12.57 ± 2.07) when compared to telomeres of fish from the earlier stages of development (Figure 2a). Q-FISH analysis of adult diploid rainbow trout showed an unexpected increase of the telomere length 18.13 ± 4.15 (Figure 2a).

In triploids, Q-FISH analysis exhibited similar telomere length in embryos (30.71 ± 7.13) and larvae (29.38 ± 6.1). Then telomere length was decreasing to reach 18.53 ± 2.99 evidenced in the juveniles and 11.38 ± 1.97 reported in subadult individuals (Figure 2b). When compared to subadult fish, adult triploids showed increased telomeres (18.13 ± 4.15) that reached similar length to that observed in juveniles (Figure 2b).

Results of ANCOVA showed that age together with interaction between ploidy and age affected telomere length (*p* < 0.05). However, ploidy itself did not show correlation with the telomere dynamics. Post hoc analysis exhibited statistically significant difference in the telomere length between subadults and embryos, subadults and larvae, subadults and adult fish (*p* < 0.05).

### 3.3. Correlation between Body Weight and Length and Telomere Length-Related Fluorescence

Statistical analysis showed substantial correlation between telomere length and body length (*r*^2^ = 0.69, *p* > 0.05) and between telomere length and body weight (*r*^2^ = 0.52, *p* > 0.05) in the triploid rainbow trout (Figure 3 and Figure 4). For diploid fish, these correlations were lower and not statistically significant (*r*^2^ = 0.07, *r*^2^= 0.34, respectively) (Figure 3 and Figure 4). Considering age categories, the biggest correlation (but not statistically significant) between telomere length and body size and weight was found in one-year-old diploids (*r*^2^ = 0.79, *r*^2^ = 84).

## 4. Discussion

Although chromosomal distribution of telomeric repeats has been studied in many salmonids [34,35,36,37], there are only few published data concerning dynamics of telomeric DNA in salmon and trout [38,39,40]. In this research we reported age-related decrease and increase of telomere length in the diploid and triploid rainbow trout. Lengths of telomeres in the rainbow trout embryos, larvae and juveniles were similar. Significantly shortened telomeres were noticed in the subadult individuals during the second year of life (Figure 1). Unexpectedly, after substantial telomere loss observed in the two-year-old rainbow trout, telomere length in the three-year-old diploids and triploids was increased to reach the size observed in the one-year-old juveniles (Figure 1).

As similar patterns of telomere-shortening, and lengthening were observed in the fertile and sterile fish it may be presumed that processes related to reproduction do not affect dynamics of telomeres in the rainbow trout. Nevertheless, maybe the same dynamics of the telomere length should not be surprising as except issues related to sterility such as reduced gonads and decreased levels of gonadotropins, sex steroids and vitellogenin, triploid salmonid are frequently observed to be morphologically and physiologically similar to the diploid individuals [25,26]. Although, triploid fish should grow faster than diploids due to the cell size, increased heterozygosity and sterility, most observations confirmed that triploid salmonids during juvenile stage usually show equal or even worse growth than diploids and only after maturation the growth is enhanced in the triploid individuals [25]. Till juvenile stage, triploid and diploid rainbow trout have been found to have comparable energy expenditure, oxidation of glucose and amino acids or excretion rate [26,41,42,43]. In chinook salmon, no differences in the gene expression was observed between ploidies even though triploid individuals were characterized by the lower growth rate [44]. Some studies evidenced similar reaction for the environmental conditions in fish with different ploidy [45]. Triploid and diploid Atlantic salmon have similar physiological stress response including comparable expression of the oxidative stress genes [46].

Decline in telomere length observed in rainbow trout in the second year of life may be consequence of a fast growth that is specific for the premature phase in fish. Results of studies performed on both endotherms and ectotherms including fish evidenced that pace of telomere attrition is the highest during early life stages characterized by the rapid growth when cell division rate is increased [21]. Experimentally enhanced growth rate observed in the transgenic salmon was followed by much higher telomere shortening when compared to their non-manipulated siblings [40]. Interestingly, only in triploid rainbow trout correlation between telomere attrition and body size (length and weight) was observed. However, triploid specimens in this experiment were bigger than their diploid counterparts at every developmental stage from hatching till subadult stage what could have affected dynamics of the telomere change (Table 1). Results of studies concerning relationship between telomere length and animal body size are inconsistent. In humans, wild house sparrows and American Alligator increased body size is associated with the reduced telomere length [13,47]. On the contrary, no significant correlation between telomere loss rate and change in the body mass was observed in long-lived barnacle goose [48] and red-sided garter snakes [49].

Telomere shortening observed in the subadults has not been compensated by telomerase despite its high activity confirmed in rainbow trout various organs including kidney irrespective of the fish age [8]. This may suggest that level of telomerase was not sufficient to prevent growth-related telomere attrition. However, after significant telomere loss observed in the two-year-old rainbow trout, telomere length-related fluorescence in the three-year-old diploids and triploids was increased to reach the size observed in the one-year-old juveniles (Figure 1). It is not a first case when an increase of the telomere length occurs after age-related shortening. In medaka, telomeres shorten during rapid growth observed at the early stages of development, lengthen during adolescence when the growth rate slows down and shorten again what is correlated with the reduced growth [50]. In turn, in zebrafish, the telomere length increases earlier, from larvae to the adult fish then stabilizes and eventually declines substantially [20]. The mechanism responsible for the telomere elongation in the adult rainbow trout is unclear though, it can be considered that telomerase that was not able to maintain length of the telomeres during period of the rapid growth enabled not only compensation for the telomere attrition, but increase of the telomere length in the three-year-old adults which growth rate slows down a little bit (Table 1). Moreover, it is not excluded that other cellular mechanisms such as reciprocal recombination and transposition of the chromosomal terminal elements [51] may have been also utilized for the rainbow trout telomere elongation in the third year of life. In cells with knocked down telomerase reverse transcriptase (TERT) subunit, telomeres shortening was limited by independent of telomerase mechanism termed alternative lengthening of telomeres (ALT) [52].

## 5. Conclusions

Q-FISH analysis was used to study dynamics of telomere length in diploid and triploid rainbow trout with aging. Behavior of telomeres in diploid and triploid trout showed similar patterns that included decline of telomeres observed in the two-year-old individuals followed by increase of telomeres reported in three-year-old adults. The reduction of telomere length observed to occur within first two years after hatching was paralleled with the period of the fast growth. Similar pattern of the telomere dynamics observed in the diploid and sterile triploid indicated processes related to reproduction did not affect telomere dynamics in rainbow trout.

## Figures and Tables

**Figure 1 genes-11-00786-f001:**
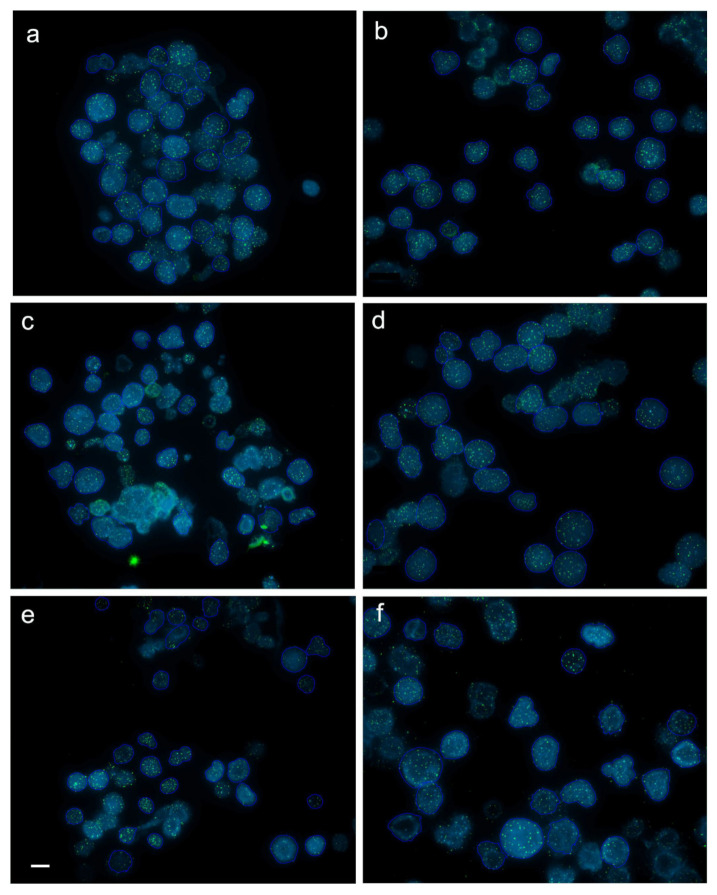
Captured images (frames) with interphase cells from juvenile (**a**,**b**), subadult (**c**,**d**) and adult (**e**,**f**) diploid (**a**,**c**,**e**) and triploid (**b**,**d**,**f**) rainbow trout after fluorescence in situ hybridization (FISH) with PNA (peptide nucleic acid) telomere probe. Average intensity of telomere fluorescence (×10) for cell from the captured frames in juveniles, subadults and adults equaled 18.8 ± 3.3 (**a**) and 18.3 ± 1.8 (**b**), 13.7 ± 1.4 (**c**) and 13.1 ± 1.4 (**d**), 26.3 ± 5.5 (**e**) and 19.3 ± 5.6 (**f**), respectively. Scale bar = 10 µm.

**Figure 2 genes-11-00786-f002:**
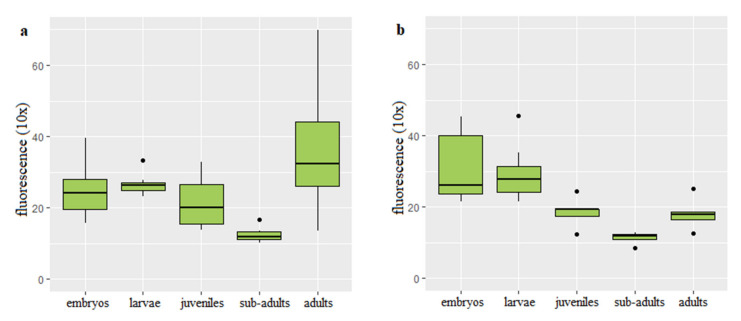
Telomere length-related fluorescence reported in diploid (**a**) and triploid (**b**) rainbow trout at different stages of development.

**Figure 3 genes-11-00786-f003:**
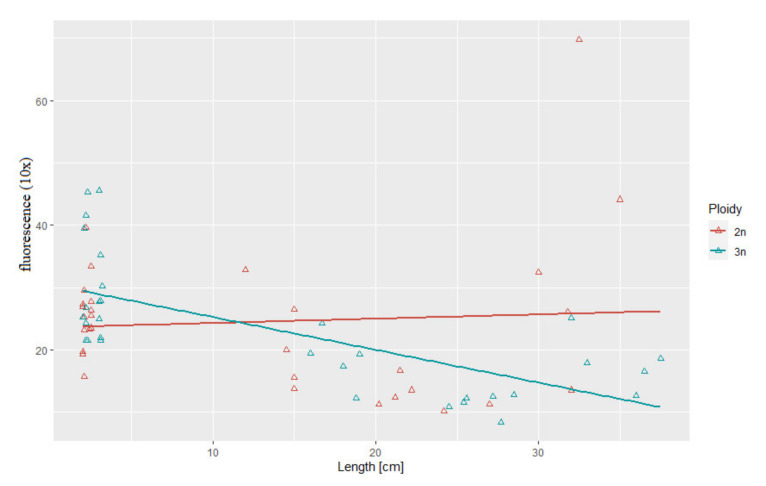
Relationship between telomere length-related fluorescence and body length in the examined triploid (*r*^2^ = 0.69, *p* > 0.05) and diploid (*r*^2^ = 0.07) rainbow trout.

**Figure 4 genes-11-00786-f004:**
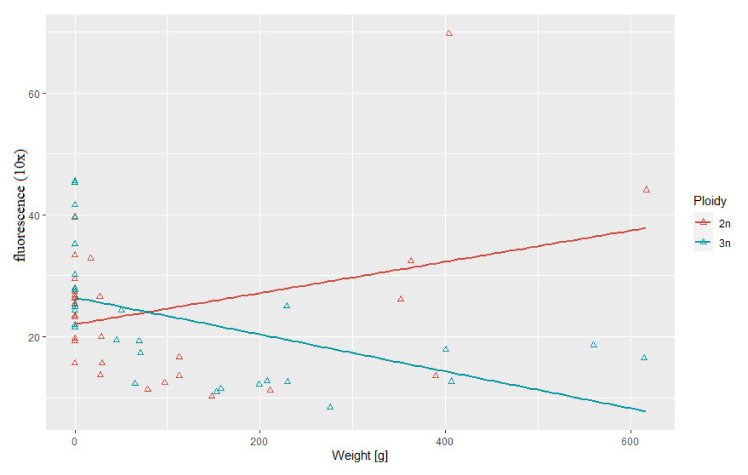
Relationship between telomere length-related fluorescence and body weight observed in examined triploid (*r*^2^ = 0.52, *p* > 0.05) and diploid (*r*^2^ = 0.34) rainbow trout.

**Table 1 genes-11-00786-t001:** Parameters of rainbow trout body weight and length. (*) indicate statistically significant difference between diploid and triploid fish at the same stage of development.

Stage of Development	Ploidy	Length (cm)		Weight (g)	
		Mean	±SD	Mean	±SD
Embryos	2n	2.08 *	0.07	0.12	0.009
	3n	2.19 *	0.10	0.12	0.010
Larvae	2n	2.43 *	3.14	0.12 *	0.017
	3n	3.08 *	0.07	0.21 *	0.021
Juveniles (one year old)	2n	14.30 *	1.30	26.12 *	5.01
	3n	17.70 *	1.31	60.28 *	11.80
Subadults (two years old)	2n	22.72 *	2.49	126.83 *	47.10
	3n	26.48 *	1.55	204.00 *	46.08
Adults (three years old)	2n	32.26	1.80	425.20	109.20
	3n	35.00	2.37	442.40	151.76

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
