# Peer review of "Telomere Dynamics in the Diploid and Triploid Rainbow Trout (Oncorhynchus mykiss) Assessed by Q-FISH Analysis"

_genes, 2020, doi:10.3390/genes11070786_

Round 1

Reviewer 1 Report

The manuscript (ID: gene-841612) titled “Telomere dynamics in the diploid and triploid rainbow trout (Oncorhynchus mykiss) assessed by Q-FISH analysis”, submitted to GENE by Panasiak et al. for a possible publication, deepens the knowledge on age- related telomere shortening in the commercially important rainbow trout Oncorhynchus mykiss, and could be useful as base information in aquaculture actions. In my opinion, the paper should be accepted only after major revisions of the English style, because it is not clear in many part, especially the discussion section. The english style should be careful cecked throughout the manuscript and revised. I suggest contacting an editing and proofreading service.

Follow detailed comments:

Results section, lines 157-170: the authors should better explained the different values reported here, in figure 1 and in table S1.

Discussion: this part is difficult to follow even after some corrections of the English style, but since I'm not a native English speaker, I think this part must be deeply revised so that it is clearer in the concepts that the authors want to express.

Table S1: I don’t understand the numbers in the column –individuals-, especially the last five. All the columns should be well aligned.

Fig. 1 caption: the last sentence should be moved to the Results section. the caption of the figure should contain only an explanation of the figure

Fig. 2 caption: the meaning of a, b, c, should be better explained both in the caption and in the results section

Other specific comment can be traced in the pdf of the manuscript.

Author Response

Reviewer 1

Comments and Suggestions for Authors

The manuscript (ID: gene-841612) titled “Telomere dynamics in the diploid and triploid rainbow trout (Oncorhynchus mykiss) assessed by Q-FISH analysis”, submitted to GENE by Panasiak et al. for a possible publication, deepens the knowledge on age- related telomere shortening in the commercially important rainbow trout Oncorhynchus mykiss, and could be useful as base information in aquaculture actions. In my opinion, the paper should be accepted only after major revisions of the English style, because it is not clear in many part, especially the discussion section. The english style should be careful cecked throughout the manuscript and revised. I suggest contacting an editing and proofreading service.

Konrad Ocalewicz (KO) – the entire manuscript has been revised and language corrected. Changes made in the ms are in different colors (new paragraphs) or observed as track changes function was ON.

Follow detailed comments:

Results section, lines 157-170: the authors should better explained the different values reported here, in figure 1 and in table S1.

KO: New figure 1 caption and detailed description of Q-FISH (l. 160-172) have been provided to explain this issue.

Discussion: this part is difficult to follow even after some corrections of the English style, but since I'm not a native English speaker, I think this part must be deeply revised so that it is clearer in the concepts that the authors want to express.

KO: Discussion part of ms has been fully revised. All major changes are seen in different colors/track changes was on when minor corrections were made.

Table S1: I don’t understand the numbers in the column –individuals-, especially the last five. All the columns should be well aligned.

KO: these odd numbers were replace by symbols of examined individuals.

Fig. 1 caption: the last sentence should be moved to the Results section. the caption of the figure should contain only an explanation of the figure

KO: the caption for this figure was full revised.

Fig. 2 caption: the meaning of a, b, c, should be better explained both in the caption and in the results section

 KO: The new statistical analysis has been performed (l. 173-182, 213-225). Results are presented in the paragraph – ll. 212-224) and in the form of new Figures (2, 3, 4)

Other specific comment can be traced in the pdf of the manuscript.

KO: all changes made are observed in different colors.

Reviewer 2 Report

Review trout telomere dynamics: triploid vs diploid

This study compares telomere length between 2n and 3n trout that exhibit very different growth rates and investment into reproductive organs. This comparison is informative because there is a natural tradeoff between investment into growth versus reproduction, and if one aspect is neglected altogether, as is the case in 3n females, then growth might not come at a cost in telomere attrition. However, the authors find that it is the 3n fish that seem to suffer telomere attrition. But this attrition comes early in life presumably (? Not explained) before investment in sexual traits might be relevant. I am curious how sex steroids (presumably lower in 3n females than 2n females) might play a role here, but is not discussed as estrogens have been suggested to play an (indirect) antioxidant role as does vitellogenin which responds to estrogen. Overall the incorporation of these traits (2n v 3n) into the introduction and discussion is critical for generating the predictions.

The manuscript is difficult to follow in places because of grammar and word choice issues. The methods need more explanation to evaluate them thoroughly. The statistical analysis is inadequate and possibly inappropriate (non-normality and unequal variance?), but no tests of these atributes are presented. The discussion takes three paragraphs to state the results and is difficult to follow and possibly does not do the data/results justice. It takes until late in the discussion to address the essential aspects of the study as it relates to the differences between 2n and 3n fish.

In the abstract, mention that this is a cross-sectional study and add at least one sentence describing the key life-history differences between triploid and diploid trout (i.e., 3n = faster growth and no or little reproductive investment). This is really an exciting comparison to make due to these differences, and it deserves more detail and should be main more explicit in the introduction and in future directions.  

I like this paper and study and think it may be worthy of publication after significant revisions.

Line-by-line comments

LL 13: “behaviour”? telomere dynamics? Or telomere length? I know Anchelin et al 2011 use “behaviour” in their title, but it still does not sound correct.

LL40; You are talking about ectotherms and growth has missed several major reviews on telomeres with and references therein:  https://royalsocietypublishing.org/toc/rstb/2018/373/1741

In particular, at least these references should be read, cited, and mined for concepts to address in this paper.

Growth

Monaghan P, Ozanne SE (2018) Somatic growth and telomere dynamics in vertebrates: relationships, mechanisms and consequences. Philosophical Transactions of the Royal Society B: Biological Sciences 373

Ectotherms

Olsson M, Wapstra E, Friesen C (2018) Ectothermic telomeres: it's time they came in from the cold. Philosophical Transactions of the Royal Society B: Biological Sciences 373

Olsson M, Wapstra E, Friesen CR (2017) Evolutionary ecology of telomeres: a review. Ann N Y Acad Sci

Vitellogenin as an antioxidant and estrogen and an antioxidant inducing agent

Lindsay WR, Friesen CR, Sihlbom C, Bergström J, Berger E, Wilson MR, Olsson M (2020) Vitellogenin offsets oxidative costs of reproduction in female painted dragon lizards. The Journal of Experimental Biology:jeb.221630

Viña J, Borrás C, Gambini J, Sastre J, Pallardó FV (2005) Why females live longer than males? Importance of the upregulation of longevity-associated genes by oestrogenic compounds. FEBS Lett 12:2541-254

Cited in: Barrett ELB, Richardson DS (2011) Sex differences in telomeres and lifespan. Aging Cell 10:913-92

I also note that most of the references are pre 2012 (with a few relevant exceptions) cite more

LL42: are equivocal (avoid unnecessary double negative)

LL 63-65: would it be clearer to write that the triploid fish continue to grow, whereas the normal diploid fish suffer from declines in growth and survival.

LL67: “which”?

LL68: odd phrasing

LL 78-80

More information about triploid sertility versus normal 2n condition, e.g., do they still invest in gonads at all (seems not)?

LL 87-95: better describe conditions, feeding regimes, lighting, outdoors/indoors? Temperatures and ranges. Etc. How were they euthanized etc.

LL 88: define dpf at 1st use

LL92 No fish older than three years were available.

LL 97-98: If this statement is meant to justify your sample size, say so and point the effect sizes in paper #12 to support that you might see differences if there are differences based on the sample size.

LL104: *briefly*

LL 110: justify head kidney (~adrenal gland) why not liver? Or heart? Organs more closely associated with health and growth?

LL 118: placing one drop

LL 125 and throughout: microscope slide not microscopic slide

LL 126: spell out fifteen.

LL 137: fix “ and what is “great truth”?,  delete.

LL137-140: The ANOVA and correlation test should be combined into a single Analysis of Covariance (ANCOVA) to test for the effects of ploidy + age class + length and body mass + interaction between age * length or mass and ploidy * life stage on telomere length. Also, there needs to be justification/verification that the distribution and variance meet model assumptions. Because body length and weight are correlated, two separate ANCOVAs should be run or use body length + residual body mass from a mass~body length regression (aka body condition) to remove the multicollinearity between “mass” at a given length and body length so that both measures of size (mass and length) can be validity included in the same model. GLM approaches with REML parameter estimates, and robust covariance estimates are probably more powerful than the classical stats used here, given the very small sample sizes.

LL138; add space

LL 149-151: it is not clear what “oocytes nor properly ovaries were observed” means, and there needs to be more detail about this in the methods and here in the results. Justify why this is reported.

LL 158: mean +/- se or sd, remind us at your first use in the results and include units.

LL 171: length is misspelled. * = statistically different a alpha = 0.05? or were post-hoc corrections applied (e.g., Tukey)?

LL 174: Figure 1 perhaps it would be more informative to include pictures (cropped and enlarged or at higher magnification to see the chromones at interphase. Also, in the methods report and reassure us that the conditions under which the images were analyzed were identical among all photos or provide some measure of variance and controls against which the images were standardized.

 LL180-182: Looking at these boxplots, it is doubtful that these data pass an equal variance test. Non-parametric analysis or data transformation is required. What is curious is that the variance is so inflated in the 3 yo 2n fish versus the 3n fish. Could there be a difference in a threshold point (or lack of one in one but not the other) for apoptosis in the two 2n v 3n? It would be nice to see the data plotted as a function of body length of condition as well.

Also, comparisons post hoc analysis of differences between telos 2n v 3n within each life stage would be useful (i.e., pairwise comparisons within this interaction: ploidy * age class in a full model).

LL 186-192: TL decreased in the 2n fish, but not 3n fish. Test for and address differences in variance between 2n & 3n at various stages. It is not clear what is meant by “3ns exhibited *higher* dynamics? Meaning 3n telomeres were shorter in the 1, 2, 3 yo than embryos and larvae? Say that.

LL 194: reported not noticed. Find an appropriate alternative word for “noticed” throughout the manuscript  e.g., observed, reported, suggested….

LL 205-208: “methabolic” speling error. Surely there is a paper that reports differences in metabolic rate between 2n and 3n?

LL 199: strain “o” delete.

Author Response

Comments and Suggestions for Authors

Review trout telomere dynamics: triploid vs diploid

This study compares telomere length between 2n and 3n trout that exhibit very different growth rates and investment into reproductive organs. This comparison is informative because there is a natural tradeoff between investment into growth versus reproduction, and if one aspect is neglected altogether, as is the case in 3n females, then growth might not come at a cost in telomere attrition. However, the authors find that it is the 3n fish that seem to suffer telomere attrition. But this attrition comes early in life presumably (? Not explained) before investment in sexual traits might be relevant. I am curious how sex steroids (presumably lower in 3n females than 2n females) might play a role here, but is not discussed as estrogens have been suggested to play an (indirect) antioxidant role as does vitellogenin which responds to estrogen. Overall the incorporation of these traits (2n v 3n) into the introduction and discussion is critical for generating the predictions.

KO: issues (sterility, hormones, growth, physiology) concerning different traits in diploid and triploid fish including trout have been incorporated in Introduction and Discussion (l. 70-75, 262-279)

The manuscript is difficult to follow in places because of grammar and word choice issues. The methods need more explanation to evaluate them thoroughly. The statistical analysis is inadequate and possibly inappropriate (non-normality and unequal variance?), but no tests of these atributes are presented. The discussion takes three paragraphs to state the results and is difficult to follow and possibly does not do the data/results justice. It takes until late in the discussion to address the essential aspects of the study as it relates to the differences between 2n and 3n fish.

KO: The new statistical analysis has been performed (l. 173-182, 213-225). Results are presented in form of new Figures (2, 3, 4)

In the abstract, mention that this is a cross-sectional study and add at least one sentence describing the key life-history differences between triploid and diploid trout (i.e., 3n = faster growth and no or little reproductive investment). This is really an exciting comparison to make due to these differences, and it deserves more detail and should be main more explicit in the introduction and in future directions.  

KO: the Abstract was reformulated according reviewer’s suggestions – we mentioned that this study was cross-sectional one (l. 14), a sentence concerning differences between 2n and 3n fish was provided (ll. 15, 16).

I like this paper and study and think it may be worthy of publication after significant revisions.

KO: Thank you very much!

Line-by-line comments

LL 13: “behaviour”? telomere dynamics? Or telomere length? I know Anchelin et al 2011 use “behaviour” in their title, but it still does not sound correct.

KO: we replaced word “behavior” with words dynamics or change or lengthening / shortening in the entire text – track changes was ON. (see Abstract, l. 85).

LL40; You are talking about ectotherms and growth has missed several major reviews on telomeres with and references therein:  https://royalsocietypublishing.org/toc/rstb/2018/373/1741

In particular, at least these references should be read, cited, and mined for concepts to address in this paper.

KO: papers concerning telomere biology in ecto and endotherms have been cited

Growth

Monaghan P, Ozanne SE (2018) Somatic growth and telomere dynamics in vertebrates: relationships, mechanisms and consequences. Philosophical Transactions of the Royal Society B: Biological Sciences 373

KO: citation no. 21 in the text and in the reference list

Ectotherms

Olsson M, Wapstra E, Friesen C (2018) Ectothermic telomeres: it's time they came in from the cold. Philosophical Transactions of the Royal Society B: Biological Sciences 373

KO: citation no. 12 in the text and in the reference list

Olsson M, Wapstra E, Friesen CR (2017) Evolutionary ecology of telomeres: a review. Ann N Y Acad Sci

KO: citation no. 12 in the text and in the reference list

Vitellogenin as an antioxidant and estrogen and an antioxidant inducing agent

Lindsay WR, Friesen CR, Sihlbom C, Bergström J, Berger E, Wilson MR, Olsson M (2020) Vitellogenin offsets oxidative costs of reproduction in female painted dragon lizards. The Journal of Experimental Biology:jeb.221630

Viña J, Borrás C, Gambini J, Sastre J, Pallardó FV (2005) Why females live longer than males? Importance of the upregulation of longevity-associated genes by oestrogenic compounds. FEBS Lett 12:2541-254

Cited in: Barrett ELB, Richardson DS (2011) Sex differences in telomeres and lifespan. Aging Cell 10:913-92

 KO: papers concerning relationship between sex, telomere, steroids have been cited : citation no. 22 23, 24,  in the text and in the reference list

I also note that most of the references are pre 2012 (with a few relevant exceptions) cite more

KO: new revies and original papers have been cited here (reference list).

LL42: are equivocal (avoid unnecessary double negative)

KO: corrected l. 53.

LL 63-65: would it be clearer to write that the triploid fish continue to grow, whereas the normal diploid fish suffer from declines in growth and survival.

KO: revised , see l. 70-74.

LL67: “which”?

KO: corrected – l. 71.

LL68: odd phrasing

KO: new sentence was provided l. 78-80.

More information about triploid sertility versus normal 2n condition, e.g., do they still invest in gonads at all (seems not)?

KO: see lines 68-76, 264-279.

LL 87-95: better describe conditions, feeding regimes, lighting, outdoors/indoors? Temperatures and ranges. Etc. How were they euthanized etc.

KO: new paragraph has been provided, l 104-117.

LL 88: define dpf at 1st use

KO: dpf was explained – l. 95

LL92 No fish older than three years were available.

KO: corrected , see line 99.

LL 97-98: If this statement is meant to justify your sample size, say so and point the effect sizes in paper #12 to support that you might see differences if there are differences based on the sample size.

KO: better description and explanation has been provided: ll. 121-125.

LL104: briefly*

KO: corrected – l. 131.

LL 110: justify head kidney (~adrenal gland) why not liver? Or heart? Organs more closely associated with health and growth?

KO: it has been justified and explain on ll. 140-142.

LL 118: placing one drop

KO: corrected l. 149

LL 125 and throughout: microscope slide not microscopic slide

KO: “microscope slide” has been used in the entir3e text – l. 149, 152.

LL 126: spell out fifteen.

KO: It has been spelled – l. 157.

LL 137: fix “ and what is “great truth”?,  delete.

KO: it has been deleted. L. 174.

LL137-140: The ANOVA and correlation test should be combined into a single Analysis of Covariance (ANCOVA) to test for the effects of ploidy + age class + length and body mass + interaction between age * length or mass and ploidy * life stage on telomere length. Also, there needs to be justification/verification that the distribution and variance meet model assumptions. Because body length and weight are correlated, two separate ANCOVAs should be run or use body length + residual body mass from a mass~body length regression (aka body condition) to remove the multicollinearity between “mass” at a given length and body length so that both measures of size (mass and length) can be validity included in the same model. GLM approaches with REML parameter estimates, and robust covariance estimates are probably more powerful than the classical stats used here, given the very small sample sizes.

The new statistical analysis has been performed (l. 173-182, 213-225). Results are presented in the text (ll. 213-225) and in the form of new Figures (3, 4)

LL138; add space

KO: the entire part has been revised.

LL 149-151: it is not clear what “oocytes nor properly ovaries were observed” means, and there needs to be more detail about this in the methods and here in the results. Justify why this is reported.

KO: better explanation has been provided, ll. 127-129.

LL 158: mean +/- se or sd, remind us at your first use in the results and include units.

KO: used in l. 203

LL 171: length is misspelled. * = statistically different a alpha = 0.05? or were post-hoc corrections applied (e.g., Tukey)?

KO: Tukey test has been performed – ll. 180/181.

LL 174: Figure 1 perhaps it would be more informative to include pictures (cropped and enlarged or at higher magnification to see the chromones at interphase. Also, in the methods report and reassure us that the conditions under which the images were analyzed were identical among all photos or provide some measure of variance and controls against which the images were standardized.

KO: such statement has been provided – l. 172, 173.

 LL180-182: Looking at these boxplots, it is doubtful that these data pass an equal variance test. Non-parametric analysis or data transformation is required. What is curious is that the variance is so inflated in the 3 yo 2n fish versus the 3n fish. Could there be a difference in a threshold point (or lack of one in one but not the other) for apoptosis in the two 2n v 3n? It would be nice to see the data plotted as a function of body length of condition as well.

KO: new figures are provided – Figure 3 and 4.

Also, comparisons post hoc analysis of differences between telos 2n v 3n within each life stage would be useful (i.e., pairwise comparisons within this interaction: ploidy * age class in a full model).

The new statistical analysis has been performed (l. 173-182, 213-225). Results are presented in the text (ll. 213-225) and in the form of new Figures (3, 4)

LL 186-192: TL decreased in the 2n fish, but not 3n fish. Test for and address differences in variance between 2n & 3n at various stages. It is not clear what is meant by “3ns exhibited *higher* dynamics? Meaning 3n telomeres were shorter in the 1, 2, 3 yo than embryos and larvae? Say that.

KO: new statistical analysis put new light on our observations – see ll. 214-226, Figure 3 and figure 4. Discussion part – ll. 289-298.

LL 194: reported not noticed. Find an appropriate alternative word for “noticed” throughout the manuscript  e.g., observed, reported, suggested….

KO: word “notice/ed” has been replaced by other words in the text.

LL 205-208: “methabolic” speling error. Surely there is a paper that reports differences in metabolic rate between 2n and 3n?

KO: spelling error has been corrected and new references provided – see references no. 25, 26, 41-46

LL 199: strain “o” delete.

KO: this word was deleted as this part of text was fully revised – 262-280

Round 2

Reviewer 2 Report

I am satisfied with how the authors' addressed my comments. I am happy to see this paper published. 

Author Response

Response to editorial notes

This revised version of the MS is substantially improved than previous one:  authors consider and respond to all requests by the reviewers. Now the MS agrees with the standard required for publication. I still suggest some minor corrections that I reported here below

Line 116 here and along the text the terms juveniles, sub-adults and adults are used. In Table 1 these terms are associated to 1-year old…etc. But in Figure 2 labels are “1-year-old”, “2-year-old” etc. Please replace (or add) the terms juveniles, sub-adults and adults.

KO: Figure 1 has been corrected according to this comment.

Line 118: Zebrafish? This probably comes from a different manuscript.

KO: Yes, this sentence has been deleted.

Line 119: “Unequal sample sizes between experiments need to be considered during discussion of the obtained results”. This likely was a suggestion by the reviewers that has been inappropriately included in the text?!

KO: This one has been deleted from the manuscript.

Line 168: add letters or use punctuation (;), otherwise the different options are not clear.

KO: This paragraph has been reworded an corrected. See ll. 166-173.

Supplementary Table 1:  take in columns so that they can fit within the page (last column is now out)

KO: Table S1 has been corrected to fit within the page.

There are still some minor type/language errors that require corrections

KO: The entire manuscript has been carefully read and corrected. Track changes has been on to follow the minor corrections.